# Peerj

# *Poppr*: an R package for genetic analysis of populations with clonal, partially clonal, and/or sexual reproduction

Zhian N. Kamvar[1], Javier F. Tabima[1] and Niklaus J. Grünwald[1,2]

[1] Department of Botany and Plant Pathology, Oregon State University, Corvallis, OR, USA
[2] Horticultural Crops Research Laboratory, USDA-ARS, Corvallis, OR, USA

## ABSTRACT

Many microbial, fungal, or oomcyete populations violate assumptions for population genetic analysis because these populations are clonal, admixed, partially clonal, and/or sexual. Furthermore, few tools exist that are specifically designed for analyzing data from clonal populations, making analysis difficult and haphazard. We developed the R package *poppr* providing unique tools for analysis of data from admixed, clonal, mixed, and/or sexual populations. Currently, *poppr* can be used for dominant/codominant and haploid/diploid genetic data. Data can be imported from several formats including *GenAlEx* formatted text files and can be analyzed on a user-defined hierarchy that includes unlimited levels of subpopulation structure and clone censoring. New functions include calculation of Bruvo's distance for microsatellites, batch-analysis of the index of association with several indices of genotypic diversity, and graphing including dendrograms with bootstrap support and minimum spanning networks. While functions for genotypic diversity and clone censoring are specific for clonal populations, several functions found in *poppr* are also valuable to analysis of any populations. A manual with documentation and examples is provided. *Poppr* is open source and major releases are available on CRAN: http://cran.r-project.org/package=poppr. More supporting documentation and tutorials can be found under 'resources' at: http://grunwaldlab.cgrb.oregonstate.edu/.

Corresponding author
Zhian N. Kamvar,
kamvarz@science.oregonstate.edu

# INTRODUCTION

The Wright–Fisher model of populations is one of the oldest models utilized in population genetic theory. Populations in this model are characterized as having non-overlapping generations with a constant size free from any selective pressures (*Weir, 1996*; *Hartl & Clark, 1997*; *Nielsen & Slatkin, 2013*). Conceptually, these populations are represented as pools of alleles that are independently assorting where random mating is approximated by randomly sampling alleles with replacement from one generation to the next. Assumptions of this model, or related models, are implicitly assumed for common population genetic analysis tools. In clonal populations, however, alleles are not independently passed on from one generation to the next, and these assumptions are violated. Classical textbooks

on population genetics do not provide much guidance on how to analyze clonal or mixed clonal and sexual populations. In reality, many populations are not strictly clonal or sexual, but can range from completely sexual to completely clonal and this is commonly observed for fungal, oomycete, or microbial populations (*Anderson & Kohn, 1995*; *Milgroom, 1996*). Currently, analysis of these populations is not straightforward as we lack the sophisticated tools and methods developed for model populations that are typically either haploid or diploid (*Grünwald & Goss, 2011*).

Inferring population structure with many commonly used model-based clustering approaches such as the program STRUCTURE (*Pritchard, Stephens & Donnelly, 2000*) is inherently problematic for clonal populations. These approaches cannot be used as clonal populations violate basic assumptions of panmixia and Hardy–Weinberg equilibrium. Thus, model free methods such as those relying on k-means clustering, dendrograms including bootstrap support for clades, or minimum spanning networks are more appropriate (*Goss et al., 2009*; *Cooke et al., 2012*; *Mascheretti et al., 2008*). Furthermore, analysis of mixed or clonal populations traditionally relies on calculation of diversity of genotypes observed and analysis of clone-censored versus non-censored populations (*McDonald, 1997*; *Milgroom, 1996*; *Grünwald & Hoheisel, 2006*). Clone censoring involves reduction of any population sample to a single observation for each multilocus genotype (MLG) in a population thereby approximating panmictic populations and removing the effect of genetic linkage (*Milgroom, 1996*). Analysis of diversity, in turn, involves calculation of the number of genotypes observed (richness), diversity, and evenness (*Grünwald et al., 2003*). Typical measures of genotypic diversity are borrowed from ecology and use either the Shannon-Wiener or Stoddart and Taylor index (*Stoddart & Taylor, 1988*; *Shannon, 1948*; *Grünwald et al., 2003*).

A critical aspect of analyzing clonal or mixed populations is testing a null hypothesis of panmixia (*Milgroom, 1996*). Testing of this hypothesis for potentially clonal populations typically relies on assessment of linkage disequilibrium among loci (*Milgroom, 1996*). This is achieved via calculation of the index of association or related indices in combination with resampling of the data to obtain a null distribution for the expectation of random mating (*Burt et al., 1996*; *Brown, Feldman & Nevo, 1980*; *Smith et al., 1993*; *Milgroom, 1996*). These approaches have, for example, been applied to *Pyrenophora teres* (*Peever & Milgroom, 1994*) and *Aphanomyces euteiches* (*Grünwald & Hoheisel, 2006*) and are routinely used in the analyses of clonal populations although they are not easily calculated given available software including MULTILOCUS, which is no longer supported, and LIAN, which only works for haploids (*Agapow & Burt, 2001*; *Haubold & Hudson, 2000*).

Hierarchical sampling adds another layer of complexity to analysis of clonal populations. With microbial populations, the geographic structure of each population is not entirely clear, and it is often important to sample temporally to see if clones persist over time (*Grünwald & Hoheisel, 2006*). A common approach when faced with multiple levels of sampling is to create a separate data set for each level or combination of levels and to analyze them separately. However, the number of data sets undergo a factorial increase with each hierarchical level, therefore increasing the chances of human error in data

reformatting or analysis. Thus, tools are needed for analysis of population data across hierarchies or subsets of data.

Here, we introduce the R package *poppr* that is specifically designed for analysis of populations that are clonal, admixed, and/or sexual. *Poppr* complements and builds on previously existing R packages including *adegenet* and *vegan* (*Jombart, 2008*; *Jombart & Ahmed, 2011*; *Oksanen et al., 2013*) while implementing tools novel to R significantly facilitating data import, population genetic analyses, and graphing of populations. These tools include among others: analysis across hierarchies of populations, subsetting of populations, clone-censoring, Bruvo's genetic distance (*Bruvo et al., 2004*), the index of association and related statistics (*Brown, Feldman & Nevo, 1980*; *Smith et al., 1993*), and bootstrap support for trees based on Bruvo's distance. By providing a centralized suite of tools appropriate for many data types, this package represents a novel and useful resource specifically tailored for analysis of clonal populations that is flexible enough for use in analysis of any population.

## MATERIALS AND METHODS

### Data import

*Poppr* allows import of data in several formats for dominant/codominant, haploid/diploid and geographic data. The R package *adegenet*, that defines the `genind` data structure that *poppr* utilizes, allows support for importing data natively from STRUCTURE, GENETIX, GENEPOP, and FSTAT. While these formats are very common and widely supported, these do not allow for import of geographic and/or regional data. Furthermore, *adegenet* will only handle diploids with this format, though manual import is possible. To aid in importing data, *poppr* has newly added the function `read.genalex()`, to read data from *GenAlEx* formatted text files into the `genind` data object of the package *adegenet* (*Jombart, 2008*; *Jombart & Ahmed, 2011*; *Peakall & Smouse, 2006*). *GenAlEx* is a popular add-in for MICROSOFT EXCEL that can handle data including codominant/dominant and haploid/diploid markers as well as geographic and regional data. This function further facilitates the import of haploid, geographic, and regional data.

Transferring data to new formats and manipulating data by hand, such as collapsing data into clones or subsetting data into different hierarchical levels, is tedious, creates redundancy, and can result in lost or misrepresented data. *Poppr* includes tools to automate such repetitive tasks. Many currently available data formats and software implementations allow analysis of only one or two levels of a population hierarchy. With *poppr* the user can import a single data set with an unlimited number of hierarchical levels. This is achieved by having the user combine the levels using a common delimiter (e.g., "Year_Country_City"). These combined levels are then used as the defining population factor in the input file and can easily be manipulated within R.

### Data analysis

Once data is imported into R, the user can dynamically access and manipulate the population hierarchy with the function `splitcombine()`, subset the data set by

population with `popsub()`, and check for cloned multilocus genotypes using `mlg()`. For data sets that include clones, the *poppr* function `clonecorrect()` will censor clones with respect to any level of a population hierarchy. In the case of missing data we use the commonly implemented, most parsimonious approach of treating missing states as novel alleles. This inherently makes analysis sensitive to missing data and genotyping error, but the user has tools available such as `missingno()` to filter out missing data at a per-individual or per-locus level. The user can also decide how uninformative loci (e.g., alleles occurring at minor frequencies; monomorphic loci; fixed heterozygous loci) are treated using the function `informloci()`. Thus, the user can specify a frequency for removal of uninformative loci. The user is encouraged to conduct analysis with and without missing data/uninformative loci to assess sensitivity to these issues when making inferences. A full list of functions available in *poppr* is provided in Table 1.

Typical analyses in *poppr* start with summary statistics for diversity, rarefaction, evenness, MLG counts, and calculation of distance measures such as Bruvo's distance, providing a suitable stepwise mutation model appropriate for microsatellite markers (*Bruvo et al., 2004*). *Poppr* will define MLGs in your data set, show where they cross populations, and can produce graphs and tables of MLGs by population that can be used for further analysis with the R package *vegan* (*Oksanen et al., 2013*). Many of the diversity indices calculated by the *vegan* function `diversity()` are useful in analyzing the diversity of partially clonal populations. For this reason, *poppr* features a quick summary table (Table 2) that incorporates these indices along with the index of association, $I_A$ (*Brown, Feldman & Nevo, 1980*; *Smith et al., 1993*), and its standardized form, $\bar{r}_d$, which accounts for the number of loci sampled (*Agapow & Burt, 2001*). Both measures of association can detect signatures of multilocus linkage and values significantly departing from the null model of no linkage among markers are detected via permutation analysis utilizing one of four algorithms described in Table 3 (*Agapow & Burt, 2001*). The user can specify the number of samples taken from the observed data set to obtain the null distribution expected for a randomly mating population. Detailed examples of these analyses can be found in the *poppr* manual.

## Visualizations

*Poppr* generates bar charts of MLG counts found within each population of your data set (Fig. 1). Histograms with rug plots for $I_A$ and $\bar{r}_d$ allow visual assessment of the quality of the distribution derived from resampling to see if a higher number of replications are necessary (Fig. 2). *Poppr* automatically produces custom minimum spanning networks for Bruvo's or other distances using Prim's algorithm, as implemented in the package *igraph* (*Csardi & Nepusz, 2006*), with the functions `bruvo.msn()` for Bruvo's distance (Fig. 3) and `poppr.msn()` for any distance matrix. The combination of data structures from *adegenet* and *igraph* allow graphing that is color coded by population with vertices grouped by MLG (*Jombart, 2008*; *Jombart & Ahmed, 2011*; *Csardi & Nepusz, 2006*). The user can further customize the appearance of the graphs directly within R by utilizing *igraph*'s `layout()` and `plot()` functions. *Poppr* also includes visualization

**Table 1** Functions found in *poppr* and their short descriptions.

| Function | Description |
| --- | --- |
| **Import/Export** | |
| getfile | Provides a quick GUI to grab files for import |
| read.genalex | Read *GenAlEx* formatted csv files to a genind object |
| genind2genalex | Converts genind objects to *GenAlEx* formatted csv files |
| **Manipulation** | |
| missingno | Handles missing data |
| clonecorrect | Clone censors at a specified population hierarchy |
| informloci | Detects and removes phylogenetically uninformative loci |
| popsub | Subsets genind objects by population |
| shufflepop | Shuffles genotypes at each locus using four different shuffling algorithms (details in Table 3) |
| splitcombine | Manipulates population hierarchy |
| **Analysis** | |
| bruvo.boot | Produces dendrograms with bootstrap support based on Bruvo's distance |
| bruvo.dist | Calculates Bruvo's distance |
| diss.dist | Calculates the percent allelic dissimilarity |
| ia | Calculates the index of association |
| mlg | Calculates the number of multilocus genotypes |
| mlg.crosspop | Finds all multilocus genotypes that cross populations |
| mlg.table | Returns a table of populations by multilocus genotypes |
| mlg.vector | Returns a vector of a numeric multilocus genotype assignment for each individual |
| poppr | Returns a diversity table by population |
| poppr.all | Returns a diversity table by population for all compatible files specified |
| **Visualization** | |
| greycurve | Helper to determine the appropriate parameters for adjusting the grey level for msn functions |
| bruvo.msn | Produces minimum spanning networks based off Bruvo's distance colored by population |
| poppr.msn | Produces a minimum spanning network for any pairwise distance matrix related to the data |

**Table 2** **Summary table produced by the** poppr() **function.** Table shown as it would appear in the R console produced by the poppr() function with 999 permutations to calculate $I_A$ and $\bar{r}_d$ *p*-values from the Aeut data set in *poppr* from *Grünwald et al. (2003)*. Table was obtained with the following code: library(poppr); data(Aeut); poppr(Aeut, sample = 999).

| Pop | N | MLG | eMLG | SE | H | G | Hexp | E.5 | Ia | p.Ia | rbarD | p.rD |
| --- | --- | --- | --- | --- | --- | --- | --- | --- | --- | --- | --- | --- |
| Athena | 97 | 70 | 65.981 | 1.246 | 4.063 | 42.193 | 0.986 | 0.721 | 2.906 | 0.001 | 0.072 | 0.001 |
| Mt. Vernon | 90 | 50 | 50.000 | 0.000 | 3.668 | 28.723 | 0.976 | 0.726 | 13.302 | 0.001 | 0.282 | 0.001 |
| Total | 187 | 119 | 68.453 | 2.989 | 4.558 | 68.972 | 0.991 | 0.720 | 14.371 | 0.001 | 0.271 | 0.001 |

**Notes.**

N, census size; MLG, multilocus genotypes; eMLG, expected MLG based on rarefaction; SE, standard error from rarefaction; H, Shannon-Wiener Index; G, Stoddart and Taylor's Index; Hexp, (*Nei, 1978*) Expected Heterozygosity; E.5, Evenness ($E_5$); Ia, $I_A$; p.Ia, *p*-value for $I_A$; rbarD, $\bar{r}_d$; p.rD, *p*-value for $\bar{r}_d$.

**Table 3 Permutation algorithms in *poppr*.** These are implemented in the calculation of $I_A$ and $\bar{r}_d$ $p$-values iterated over all loci independently.

| Method | Name | Units sampled | With replacement | Weight |
|---|---|---|---|---|
| 1 | permutation | alleles | No | - |
| 2 | parametric bootstrap | alleles | Yes | allele frequencies |
| 3 | non-parametric bootstrap | alleles | Yes | equal |
| 4 | multilocus | genotypes | No | - |

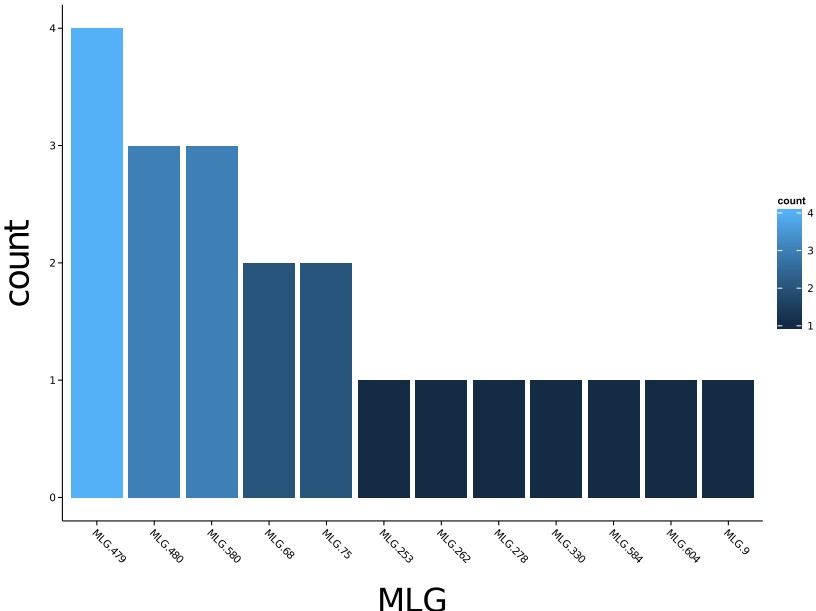

**Figure 1 Multilocus genotype histogram.** Distribution of 12 multilocus genotypes from the Finland population of the H3N2 SNP data set (*Jombart, 2008*).

of dendrograms using UPGMA (*Schliep, 2011*) and Neighbor-Joining (*Paradis, Claude & Strimmer, 2004*) algorithms with bootstrap support for Bruvo's distance using the function `bruvo.boot()` (Fig. 4). Neither graphing of minimum spanning networks or dendrograms with bootstrap support are currently possible for populations in any other R packages.

## Performance

Most of the functions in *Poppr* were written and optimized for performance in R and are available for inspection and/or download at https://github.com/grunwaldlab/poppr. Algorithms of $\geq O(n^2)$ complexity were written in the byte-compiled C language to optimize runtime performance.

For comparisons of $I_A$ and Bruvo's distance, we utilized the data set `nancycats` (237 diploid individuals genotyped at nine microsatellite loci) from the *adegenet* package. Calculations were run independently 10 times and then averaged. Bruvo's distance was

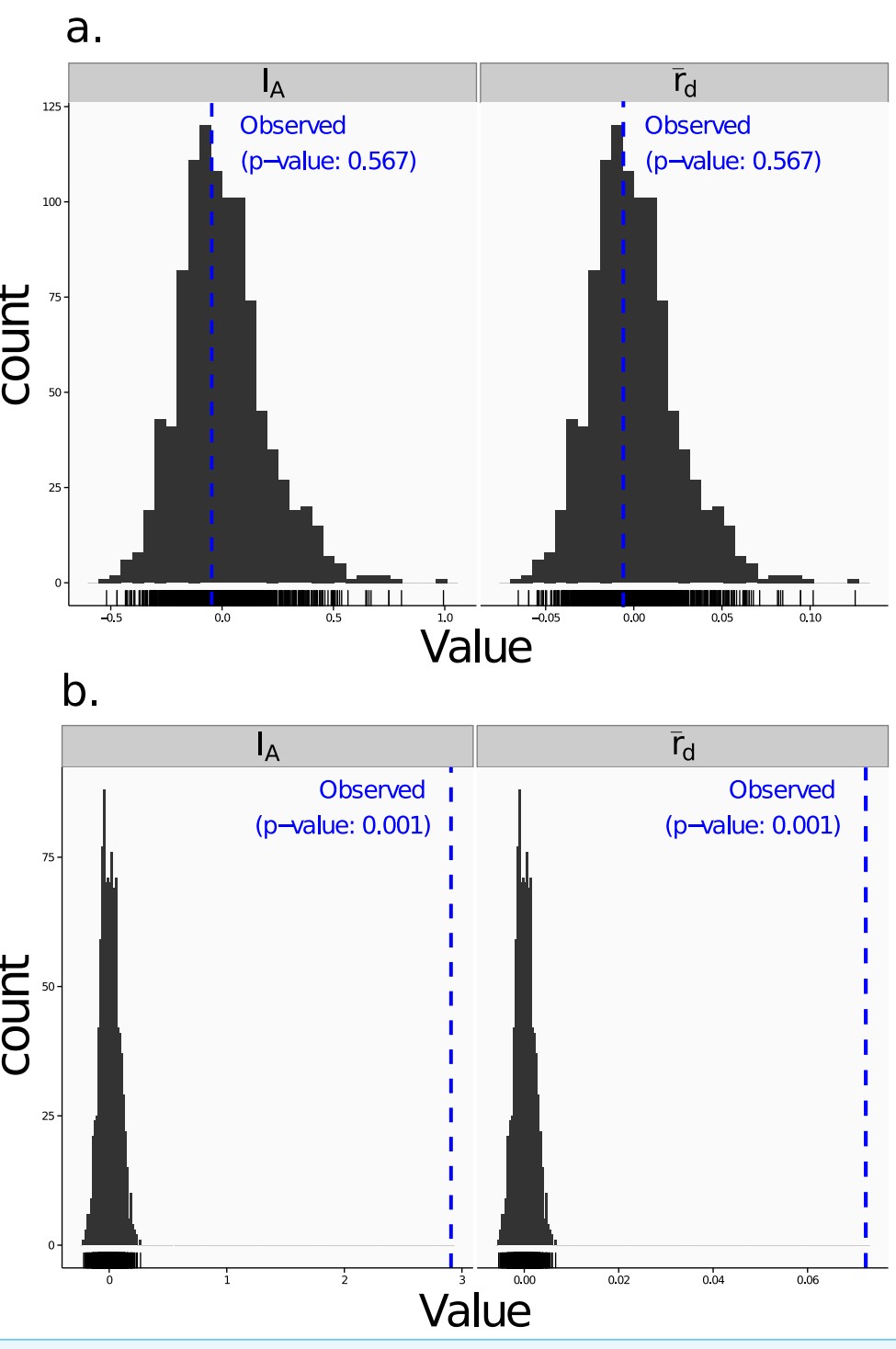

**Figure 2 Linkage disequilibrium.** Visualizations of tests for linkage disequilibrium, where observed values (blue dashed lines) of $I_A$ and $\bar{r}_d$ are compared to histograms showing results of 999 permutations using method 1 in Table 1. Results are shown for the sexual population 5 of the `nancycats` data set (*Jombart, 2008*) (A) and for the clonal `Athena` population of the `Aeut` data set (*Grünwald et al., 2003*) (B).

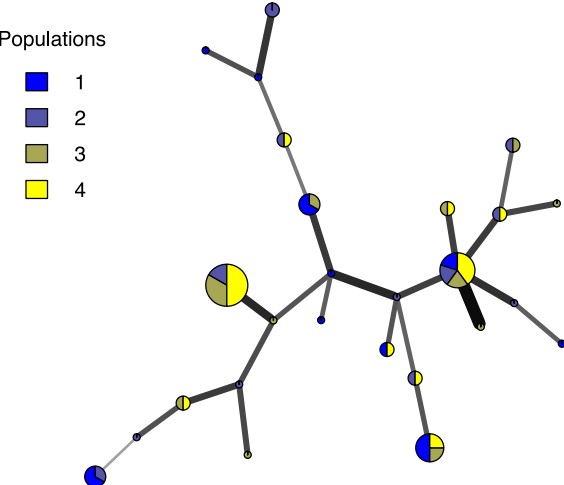

Populations
- 1
- 2
- 3
- 4

**Figure 3 Minimum spanning network.** Example minimum spanning network using Bruvo's distance on a simulated partially clonal data set with 50 individuals genotyped over 10 microsatellite loci produced with the software SimuPOP v.1.0.8 (*Peng & Amos, 2008*). Each node represents a unique multilocus genotype. Node shading (colors) represent population membership, while edge widths and shading represent relatedness. Edge length is arbitrary.

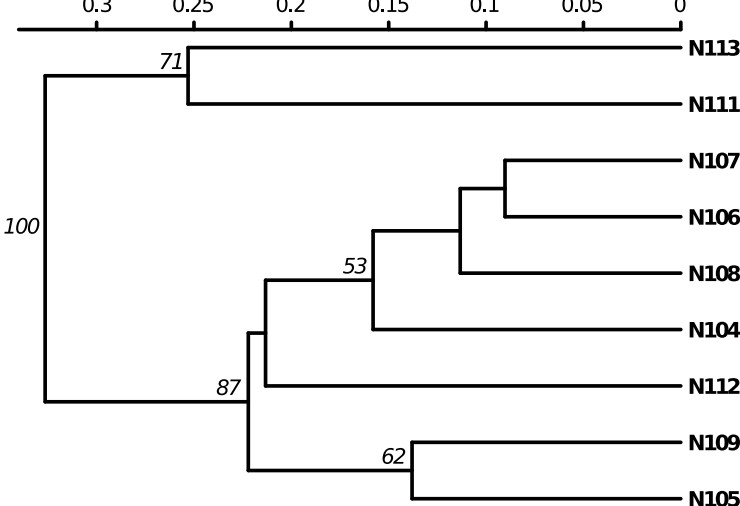

**Figure 4 Dendrogram based on genetic distance.** UPGMA tree produced from Bruvo's distance with 1000 bootstrap replicates (node values greater than 50% are shown). Data from population 9 of the `nancycats` data set (*Jombart, 2008*).

calculated on a machine with OSX 10.8.4 and a 2.9 GHz intel processor. The $I_A$ and $\bar{r}_d$ calculations were performed on a machine with OSX 10.5.8 and a 2.4 GHz intel processor due to the inability of the software MULTILOCUS to work on any later version of OSX.

## Example data sets

Along with example data sets preloaded into the *adegenet* package, *poppr* offers two data sets of clonal populations. The data set `Aeut` is comprised of 187 isolates of

**Table 4  Citation of methods and indices implemented in *poppr*.**

| Method/Index | Citation | Function(s) in *poppr* |
|---|---|---|
| Expected MLG (rarefaction) | *Hurlbert (1971)*; *Heck, Belle & Simberloff (1975)* (for std. err.) | `poppr()` |
| $H$ | *Shannon (1948)* | `poppr()` |
| $G$ | *Stoddart & Taylor (1988)* | `poppr()` |
| $H_{exp}$ | *Nei (1978)* | `poppr()` |
| $E_5$ | *Grünwald & Hoheisel (2006)*; *Pielou (1975)*; *Ludwig & Reynolds (1988)* | `poppr()` |
| $I_A/\bar{r}_d$ | *Brown, Feldman & Nevo (1980)*; *Smith et al. (1993)* ($I_A$), *Agapow & Burt (2001)* ($\bar{r}_d$) | `ia()` `poppr()` |
| Clone correction | *Milgroom (1996)*; *Grünwald et al. (2003)*; *Grünwald & Hoheisel (2006)* | `clonecorrect()` `poppr()` |
| Minimum Spanning Networks | *Csardi & Nepusz (2006)* | `poppr.msn()` `bruvo.msn()` |
| Bruvo's Distance | *Bruvo et al. (2004)* | `bruvo.dist()` `bruvo.msn()` `bruvo.boot()` |
| Bootstrapping | *Paradis, Claude & Strimmer (2004)* | `bruvo.boot()` |
| Neighbor Joining | *Paradis, Claude & Strimmer (2004)* | `bruvo.boot()` |
| UPGMA | *Schliep (2011)* | `bruvo.boot()` |

*Aphanomyces euteiches* genotyped over 56 AFLP loci with defined populations and subpopulations (*Grünwald & Hoheisel, 2006*). The `partial_clone` data set contains 50 simulated individuals genotyped over 10 microsatellite loci produced with the software SiмuPOP v.1.0.8 (*Peng & Amos, 2008*). Each data set can be loaded into R using the commands `data(Aeut)` and `data(partial_clone)`, respectively.

### Citation of methods implemented in *poppr*

Several of the methods implemented in *poppr* are described elsewhere. Users should refer to the original publications for interpretations and citation. See Table 4 for a full list of citations. As with any R package, users should always cite the *R Core Team (2013)*.

## RESULTS AND DISCUSSION

*Poppr* provides significant, convenient tools for analysis of clonal, partially clonal, and sexual populations available in one environment on all major operating systems. The ability to analyze data for multiple populations across a user-defined hierarchy and clone-censoring provide novel functionality in R. Combined with R's graphing abilities, publication-ready figures are thus obtained conveniently.

### New functionalities

*Poppr* implements several new functionalities. As of this writing, aside from *poppr*, there exist two programs that calculate $I_A$: LIAN (*Haubold & Hudson, 2000*) and мultilocus (*Agapow & Burt, 2001*). LIAN can calculate $I_A$ for haploid data and is only available online or for ∗nix systems with a C compiler such as OSX and Linux (*Haubold & Hudson, 2000*). Мultilocus implemented $\bar{r}_d$, a novel correction for $I_A$, but is no longer

**Table 5**  Comparison of programs that calculate $I_A$.

|  | Haploids | Diploids | $\bar{r}_d$ | All Platforms | Batch Analysis |
|---|---|---|---|---|---|
| *poppr* | Yes | Yes | Yes | Yes | Yes |
| *LIAN* | Yes | No | No | Yes | Yes |
| *multilocus* | Yes | Yes | Yes | No | No |

supported (*Agapow & Burt, 2001*). MULTILOCUS will only calculate index values for one data set at a time and LIAN requires the user to structure the data set with populations in contiguous blocks to analyze multiple populations within a single file. Thus *poppr* provides significant improvements for calculation of linkage disequilibrium, and handles both haploid and diploid data, works on all major operating systems, and is capable of batch analysis of multiple files and multiple populations defined within a file including the possibility of clone correction and sub-setting. A comparison of the capabilities of these programs are summarized in Table 5.

To test significance for $I_A$ and $\bar{r}_d$, *poppr* offers four permutation algorithms. Each one will randomly shuffle data at each locus, effectively unlinking the loci. The algorithm previously utilized by MUTLILOCUS is included. The MULTILOCUS-style algorithm shuffles genotypes, maintaining the associations between alleles at each locus (*Agapow & Burt, 2001*). More appropriately, alleles are expected to assort independently in panmictic populations. *Poppr* thus provides three new algorithms for permutation that allow for independent allele assortment at each locus. The default algorithm permutes the alleles at each locus and the remaining two will randomly sample alleles from a multinomial distribution parametrically and non-parametrically (*Weir, 1996*). Details of these algorithms are presented in Table 3. Because the index of association is calculated using a binary measure of dissimilarity, we have also made this available as a distance measure called `diss.dist()`. This pairwise distance is based on the percent allelic differences.

*Poppr* also newly implements Bruvo's genetic distance that utilizes a stepwise mutation model appropriate for microsatellite data (*Bruvo et al., 2004*). While this distance is implemented in the program GENODIVE (*Meirmans & Van Tienderen, 2004*) and the R package *polysat* (*Clark & Jasieniuk, 2011*), there are a few caveats with these two implementations. GENODIVE is closed-source, and only implemented in OSX. Both *poppr* and *polysat* are open-source and available on all platforms, but *polysat*, being optimized for polyploid individuals with ambiguous allelic dosage, is inappropriate for analyzing diploids. *Polysat* will collapse homozygous individuals into a single allele and attempt to infer the second allelic state in comparison with heterozygous individuals. Since haploid and diploid individuals show clear allelic dosage, this procedure creates a bias misrepresenting the true distance. Not only is *poppr* not subject to this bias, but it also newly introduces bootstrap support for this distance as shown in Fig. 4.

## Performance

*Poppr* reduces the amount of intermediate files and repetitive tasks needed for basic population genetic analyses and implements computationally intensive functions, such

**Table 6 Performance comparison.** Comparison of performance on one data set of 237 individuals over nine loci. Each time point represents an average of 10 independent runs. Calculations of $I_A$ are based on 100 permutations.

|  | $I_A$ (seconds) | Bruvo's distance (seconds) |
|---|---|---|
| *poppr* | 13.4 | 0.3 |
| *polysat* | - | 58.3 |
| *multilocus* | 547.2 | - |

as Bruvo's distance and the index of association in C to improve performance. The *polysat* package calculation of Bruvo's distance took 58.3 s on average whereas *poppr*'s calculation was over 190 times faster, averaging 0.3 s (Table 6). For calculation of $I_A$ and $\bar{r}_d$ with 100 permutations and Nei's genotypic diversity (*Nei, 1978*), MULTILOCUS required around 9.12 min on average, as compared to 13.4 s with *poppr*.

## CONCLUSIONS

The R package *poppr* provides new functions and tools specifically tailored for analysis of data from clonal or partially clonal populations. No software currently available provides this set of tools. Novel capabilities include analysis across multiple populations at multiple levels of hierarchies, clone-censoring, and subsetting. These in combination with R's command line interface and scripting capabilities makes analyses of these populations more streamlined and tractable. By implementing computationally expensive algorithms such as Bruvo's distance and $I_A$ in C, analyses of multiple populations that would normally take hours to complete can now be finished in a matter of minutes. This allowed us to expand the utility of these measures to convenient new graphing abilities such as automatically creating dendrograms with bootstrap support for Bruvo's distance and minimum spanning networks. In addition to improved performance, many of the new tools in *poppr* may also be applied to non-clonal populations. While major releases of *poppr* are available on CRAN, we are continuing to develop this package to be able to efficiently handle genome-sized SNP data. Development versions are available on GitHub at https://github.com/grunwaldlab/poppr.

## ACKNOWLEDGEMENTS

The authors would like to thank Sydney Everhart and Corine Schoebel for invaluable alpha testing and Paul-Michael Agapow for providing the *multilocus* C++ source code for reference. We thank Sydney Everhart and Brian Knaus for comments that significantly improved this manuscript. We also thank the ad hoc reviewers for their valuable comments and feedback. Mention of trade names or commercial products in this manuscript are solely for the purpose of providing specific information and do not imply recommendation or endorsement.

### Funding

The work was funded in part by USDA ARS and USDA NIFA. The funders had no role in study design, data collection and analysis, decision to publish, or preparation of the manuscript.

### Grant Disclosures

The following grant information was disclosed by the authors:
USDA NIFA grant: 2011-68004-30154.
USDA ARS: 5358-22000-039-00D.

### Competing Interests

NJG is an Academic Editor for PeerJ. The authors declare no additional competing interests.

### Author Contributions

- Zhian N. Kamvar conceived and designed the experiments, performed the experiments, analyzed the data, contributed reagents/materials/analysis tools, wrote the paper, prepared figures and/or tables, reviewed drafts of the paper.
- Javier F. Tabima conceived and designed the experiments, performed the experiments, analyzed the data, contributed reagents/materials/analysis tools, prepared figures and/or tables, reviewed drafts of the paper.
- Niklaus J. Grünwald conceived and designed the experiments, analyzed the data, contributed reagents/materials/analysis tools, wrote the paper, reviewed drafts of the paper.

### Data Deposition

The following information was supplied regarding the deposition of related data:
https://github.com/grunwaldlab/poppr
http://cran.r-project.org/package=poppr
http://grunwaldlab.cgrb.oregonstate.edu/poppr-r-package-population-genetics.

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
