# Peer review of "Poppr: an R package for genetic analysis of populations with clonal, partially clonal, and/or sexual reproduction"

_PeerJ, doi:10.7717/peerj.281_

## Round 0.1 · original submission · Minor Revisions

This is an important program to analyze clonal populations and we certainly welcome such an effort. However, please follow the suggestions given by the reviewers in order to make the text clear

Reviewer 1 ·

Basic reporting

The manuscript reported the development of a new software in R package to analyze clonal population. The manuscript is well written, and I think the new software would be useful to the community who are studying the population genetics ad epidemiology of pathogens and diseases.

Experimental design

The experimental design and the illustration are relevant.

Validity of the findings

No comments.

Additional comments

I have a few simple questions:
1. "function clonecorrect() will censor exact clones with respect to any level of a population hierarchy". How this function is sensitive to missing data or genotyping error (there is no explanation about it in the text)?

2. What is the algorithm / assumption in the detection and removal of phylogenetically uninformative loci in function "informloci"? What is the definition of uninformative loci in the example data?

3. Is it possible to change the size and branching pattern of the pie charts and nodes in the "custom" minimum spanning network? e.g. The most frequent genotype will be located in the central of the diagram by default! For example, may I move it up to the top right panel?

4. Does the software have sample data set?

·

Basic reporting

No comments.

Experimental design

While I value the work done by Kamvar and collaborators to produce the R package presented in the article (and I will welcome the publication of a software note about it) I am unsure it is within the scope of the journal (http://peerj.com/about/aims-and-scope/) which states that “PeerJ only considers Research Articles”. My recommendation (minor revision) will be applicable if the editor considers the article to be within the scope of PeerJ.

Validity of the findings

No Comments.

Additional comments

Kamvar and collaborators have produced an R package (Poppr) for the analysis of population genetic data that adds new useful functions that were missing from previous R packages (such as adegenet) and other software. Most of the tools implemented in Poppr allow for the management of data [e.g. import data from file with read.genalex(); split data into subsets with popSub()] or numeric and graphical summaries of the data [e.g. calculate linkage disequilibrium indexes with ia() and plot networks with poppr.msn()] that are not specific to data from clonal or partially clonal organisms. Only a group of tools allows to manage the presence of repeated genotypes in the sample, which could be useful for clonal organisms, but also to selfing populations or for error checking (e.g. same individual genotyped twice) in any context. Thus, I find a bit misleading the way the package is presented.

I agree that many model-based statistical inference programs in populations genetics are based on either the Wright-Fisher model or the coalescent model (STRUCTURE, BEAST, IMa...) that are only applicable to haploid or sexual diploid organisms. Researchers working on partially clonal, clonal diploid and other mixed reproductive systems are in a disadvantage in that sense. However, calculation of summary statistics (genetic diversity indexes, population differentiation, genetic distances, linkage disequilibrium indexes...) does not depend on the reproductive model of the organisms. There is a plethora of population genetic programs (and some R packages) that calculate such summary statistics and these are available for researchers working on clonal organisms as well as those working on “standard” organisms. Poppr extends this range of options (which is great) to some additional summary statistics that might be more frequently used in the study of clonal organisms but is not specific to such organisms.

To sum up, it seems to me that the article oversells the applicability to organisms with clonal reproduction and at the same time might prevent the usage of Poppr by researchers working on non-clonal organisms, who might ignore the article. I suggest to revise the text to give the reader a more clear picture of the true applicability of the package.

---

## Round 0.2 · accepted · Accept

I am very impressed with the speed and careful work on your manuscript.